# Non-visual light modulates behavioral memory and gene expression in *Caenorhabditis elegans*

Zhijian Ji[1†], Bingying Wang[1†], Rashmi Chandra[1†], Junqiang Liu[1†], Supeng Yang[1,2], Yong Long[3], Michael Egan[1,2], Fujia Han[4], Han Wang[4], Noelle L'Etoile[5], Dengke K Ma[1,6,7]*

[1]Cardiovascular Research Institute, University of California, San Francisco, San Francisco, United States; [2]CVRI, Department of Molecular Cell Biology, University of California, Berkeley, Berkeley, United States; [3]Key Laboratory of Breeding Biotechnology and Sustainable Aquaculture, Institute of Hydrobiology, Chinese Academy of Sciences, Wuhan, China; [4]Department of Integrative Biology, University of Wisconsin-Madison, Madison, United States; [5]Department of Cell and Tissue Biology, University of California, San Francisco, San Francisco, United States; [6]Department of Physiology, University of California San Francisco, San Francisco, San Francisco, United States; [7]Innovative Genomics Institute, Berkeley, United States

*For correspondence:
Dengke.Ma@ucsf.edu

[†]These authors contributed equally to this work

Competing interest: The authors declare that no competing interests exist.

## eLife Assessment

This **important** study uncovers a previously unrecognized light-responsive pathway in *C. elegans* that depends on live food bacteria and is mediated by the bZIP factors ZIP-2/CEBP-2 and the cytochrome P450 enzyme, CYP-14A5. The authors show that this bacteria-linked pathway modulates long-term memory and can be harnessed as a low-cost light-inducible expression system, opening new directions for sensory biology and genetic engineering in worms. The exact means by which live bacteria modulate light signal that activates ZIP-2/CEBP-2 in the worm remains to be elucidated. The evidence supporting the pathway's role uses multiple genetic, transcriptional, and behavioral assays and is **convincing**.

**Abstract** Visible light influences a range of physiological processes, yet how animals respond to it independently of the visual system remains largely unknown. Here, we uncover a previously undescribed light-induced transcriptional pathway that modulates behavioral plasticity in *Caenorhabditis elegans*, a roundworm without eyes. We demonstrate that ambient visible light or controlled-intensity visible-spectrum LED activates an effector gene *cyp-14A5* in non-neuronal tissues through the bZIP transcription factors ZIP-2 and CEBP-2. Light induction of *cyp-14A5* is more prominent at shorter wavelengths but is independent of the known blue light receptors LITE-1 and GUR-3 in *C. elegans*. This bZIP-dependent genetic pathway in non-neuronal tissues enhances behavioral adaptability and olfactory memory, suggesting a body-brain communication axis. Furthermore, we use the light-responsive *cyp-14A5* promoter to drive ectopic gene expression, causing synthetic light-induced sleep and rapid aging phenotypes in *C. elegans*. These findings advance our understanding of light-responsive mechanisms outside the visual system and offer a new genetic tool for visible light-inducible gene expression in non-neuronal tissues.

## Introduction

Visible light is crucial for image formation and regulating various physiological processes through the visual system, yet how animals respond to ambient light independently of sight remains under-studied and likely through diverse mechanisms. Recent studies have uncovered non-visual photoreception mechanisms that modulate a range of biological processes, from circadian rhythms to stress responses and metabolic homeostasis (*Andrabi et al., 2023*; *Cronin and Johnsen, 2016*; *Do and Yau, 2010*; *Van Gelder, 2008*). These mechanisms often involve specialized light-sensitive proteins, such as opsins and cryptochromes, widely expressed in body locations, including the skin, brain, and peripheral organs. For example, mammalian melanopsin-expressing retinal ganglion cells play critical roles in systemic light responses largely independent of image formation (*Berson et al., 2002*; *Hattar et al., 2003*; *Lucas et al., 2003*; *Shi et al., 2025*). In the nematode roundworm *Caenorhabditis elegans*, blue light photoreception requires the light-activated ion channels LITE-1 and GUR-3 in specific neurons, influencing aversive behaviors and cellular physiology (*Bhatla and Horvitz, 2015*; *Edwards et al., 2008*; *Gong et al., 2016*; *Hanson et al., 2023*). Visible light irradiation can also generate photo-oxidative reactive oxygen species in diverse living organisms (*De Magalhaes Filho et al., 2018*; *Liebel et al., 2012*; *Mahmoud et al., 2010*). Despite these advances, the molecular pathways and physiological outcomes of non-visual light sensing and responses remain understudied, raising intriguing questions about the mechanistic basis and functional implications of light as an environmental cue beyond vision.

We previously studied how genes encoding cytochrome P450 (CYP) proteins respond to and mediate effects of exposure to environmental stresses in *C. elegans* (*Keller et al., 2014*; *Ma et al., 2013*). Among various transcriptional reporters we generated for CYP-encoding genes to monitor environmental regulation, we serendipitously discovered that the *cyp-14A5* promoter-driven GFP expression is particularly sensitive to ambient visible light exposure. Building upon this initial finding, we conduct transcriptome profiling to identify light-inducible genes in addition to *cyp-14A5*, determine key environmental and transcriptional regulators of *cyp-14A5*, and show that the light-inducible CYP-14A5 promotes behavioral plasticity and olfactory memory in *C. elegans*. The findings also provide a genetic tool to use the light-inducible *cyp-14A5* promoter to flexibly and ectopically drive gene expression.

## Results

### Light activates expression of *cyp-14A5* and other genes in *C. elegans*

Cytochrome P450 proteins comprise a highly conserved superfamily of heme-containing monooxygenases critical for metabolizing endogenous and xenobiotic compounds (*Denisov et al., 2005*). We constructed transcriptional reporters for genes encoding CYPs in *C. elegans* and found that *cyp-14A5p::GFP* was drastically upregulated by bright-field transmission light from a microscope inadvertently left on overnight. Using controlled light versus dark conditions, we confirmed the finding from an integrated *cyp-14A5p::GFP* reporter and observed its robust and widespread GFP expression in many tissues induced by moderate-intensity (500–3000 Lux, 16–48-hr duration) LED light exposure (*Figure 1A*). The photometric Lux range is approximately 0.1–0.60 mW/cm² in radiometric (total radiant power) metric given the spectrum of the LED light source. The level of GFP expression increased proportionally with both light intensity and duration (*Figure 1B*), the condition of which does not impact ambient temperature (*Figure 1—figure supplement 1A–D*). Other common stresses, including transient 32°C heat shock, constant 24-hr hypoxia, or starvation did not apparently induce *cyp-14A5p::GFP* (*Figure 1—figure supplement 1E–H*). Interestingly, light induction of *cyp-14A5p::GFP* appears to require live bacteria sustaining a non-starved animal physiological state (*Figure 1—figure supplement 1I and J*). We are investigating how bacteria modulate host light responses in a separate study and, in the present work, focus on the regulation and functional consequences of *cyp-14A5* in *C. elegans* upon light exposure in the presence of the OP50 bacteria as food.

To determine CYP-14A5 protein expression pattern, we constructed a translational GFP reporter (*cyp-14A5p::cyp-14A5::GFP*) and observed robust light-induced expression of CYP-14A5::GFP in many of the non-neuronal tissues, including the pharynx, hypoderm, and intestine (*Figure 1C*). The transcriptional reporter exhibited similar patterns of non-neuronal GFP induction by light (*Figure 1A*). The translational reporter displayed a CYP-14A5::GFP pattern characteristic of endoplasmic reticulum

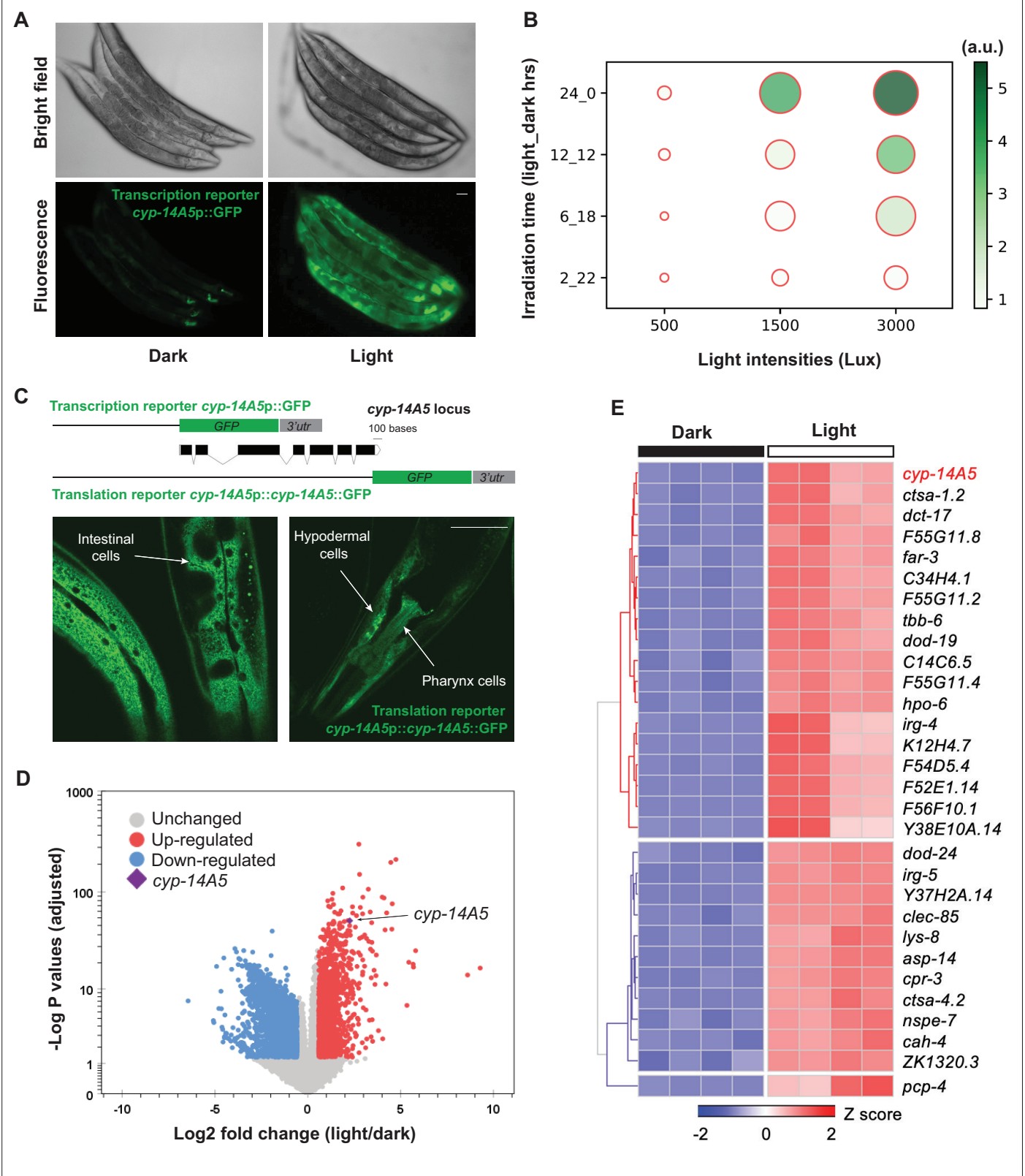

**Figure 1.** Visible light exposure activates the CYP-encoding gene *cyp-14A5* in a genetic program in *C. elegans*. (**A**) Representative epifluorescence and brightfield images showing *cyp-14A5*p::GFP induction by visible light exposure (1500 Lux, 24 hrs), in synchronized young adults (24 hrs post L4). Scale bar: 50 µm. (**B**) Bubble plot showing fold induction of *cyp-14A5p::GFP* as a function of light intensity (Lux) and duration (hours of light_dark indicated in Y axis). (**C**) Schematic of *cyp-14A5* transcriptional and translational GFP reporters. The translational reporter shows non-neuronal (indicated by arrows)

*Figure 1 continued on next page*

Figure 1 continued

induction of CYP-14A5::GFP by light (1500 Lux, 24 hrs). Scale bar: 50 μm. (**D**) Volcano plot showing genes differentially regulated by light (1500 Lux, 24 hrs), with *cyp-14A5* highlighted, in synchronized young adults (24 hrs post L4). (**E**) Heat map of top-ranking visible light-regulated genes (top 30 including *cyp-14A5*).

The online version of this article includes the following figure supplement(s) for figure 1:

**Figure supplement 1.** *cyp-14A5p::GFP* induction responds primarily to visible light exposure rather than changes in ambient oxygen, nutrient, or temperature.

**Figure supplement 2.** Gene ontology analysis of light-induced transcriptomic changes.

morphology (**Figure 1C**), consistent with the established localization of most CYP enzymes to ER membranes.

To explore how eyeless *C. elegans* responds to ambient visible light independently of a visual system, we conducted transcriptomic profiling by RNA-seq in *C. elegans* exposed to controlled light or dark conditions. We found that defined visible light exposure (1500 Lux, 24 hr duration) to a synchronized population of young adults (24 hrs post L4) triggered a robust genome-wide transcriptional response, including thousands of genes differentially regulated (adjusted p value <0.05, **Figure 1D**). Among these, *cyp-14A5* was one of the most strongly upregulated genes (**Figure 1E**). Previous work identified *gst-4* as a gene responsive to light exposure (**De Magalhaes Filho et al., 2018**). In our RNA-seq analysis, *gst-4* also was differentially regulated, and we confirmed this using a *gst-4p::GFP* reporter, although the response was relatively modest compared to the robust induction of *cyp-14A5p::GFP* by the same light condition (**Figure 1—figure supplement 1J**). Gene ontology (GO) analysis of the light-induced genes reveals their significant enrichment in several pathways, including transmembrane signaling, pathogen and stress responses, protein phosphorylation, cellular homeostasis, and metabolisms (**Figure 1—figure supplement 2**).

## ZIP-2 and CEBP-2 are essential for light-induced transcriptional responses

We next investigated the molecular regulators driving *cyp-14A5* activation in response to light. To test if it requires previously identified blue-light receptors LITE-1 or GUR-3, we crossed the *cyp-14A5p::GFP* reporter with *lite-1* and *gur-3* double loss-of-function mutants. Surprisingly, light-induced GFP expression was largely preserved in *lite-1 gur-3* double mutants, indicating that *cyp-14A5* activation operates through an alternative, non-visual light-sensing mechanism (**Figure 2A**). Prolonged photonic light exposure may also cause photo-oxidation of DNA and genotoxicity, leading to DNA damage and ATM protein-dependent checkpoints and transcriptional responses (**Ciccia and Elledge, 2010**; **De Magalhaes Filho et al., 2018**; **Schuch et al., 2017**). However, loss of the DNA damage sensor ATM-1 did not apparently affect light-induced *cyp-14A5p::GFP* (**Figure 2A**). These findings underscore the existence of a novel light-responsive pathway in *C. elegans*, distinct from previously characterized photoreceptive systems.

To identify transcriptional regulators driving *cyp-14A5* activation in response to light, we adopted an RNAi-based screening strategy, focusing initially on approximately 400 genes encoding transcription factors, including those responding to various types of stresses. Knockdown of the expression of genes encoding TF from a previously assembled RNAi library or selected for mediating various common stress responses (hypoxia, oxidative stress, heat shock, etc.) did not appear to affect *cyp-14A5* activation in response to light (**Figure 2B**). Interestingly, we identified from such screen two bZIP-type transcription factors, ZIP-2 and CEBP-2, as critical mediators of light-induced *cyp-14A5* transcription (**Figure 2B**). Knockdown or genetic ablation of either *zip-2* or *cebp-2* abolished light-induced *cyp-14A5p::GFP* expression (**Figure 2B and C**). ZIP-2 and CEBP-2 have been previously identified (**Dunbar et al., 2012**; **Estes et al., 2010**; **Kniazeva and Ruvkun, 2025**; **Reddy et al., 2016**) to cooperate in a regulatory complex and mediate transcriptional responses to translational inhibition caused by the bacterial pathogen *Pseudomonas aeruginosa* PA14. In these studies, the *irg-1*p::GFP transcriptional reporter was robustly activated by PA14 as a well-established target for ZIP-2 and CEBP-2. Interestingly, we found *irg-1*p::GFP was not activated by the same light condition (1500 Lux, 24 hrs) that reliably induced *cyp-14A5*p::GFP (**Figure 2D**). Although ZIP-2 can be activated by pathogen stresses through ribosomal inhibition and subsequent selective ZIP-2 translation (**Dunbar et al., 2012**;

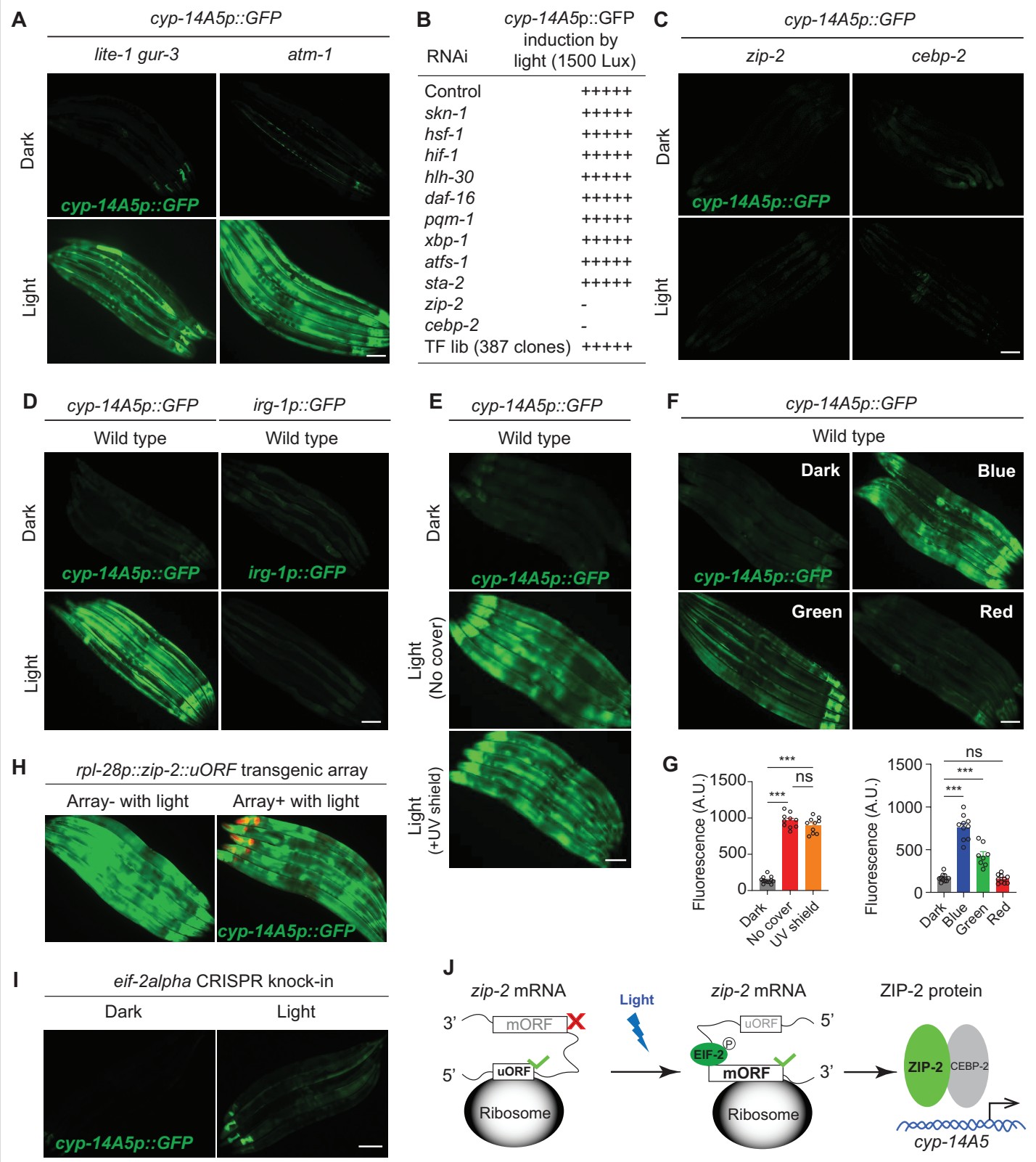

**Figure 2.** Light induction of *cyp-14A5*p::GFP requires the transcription factors ZIP-2 and CEBP-2. (**A**) Representative epifluorescence images showing light-induced *cyp-14A5*p::GFP expression in *lite-1 gur-3* double and *atm-1* single loss-of-function mutants. Scale bar: 50 μm. (**B**) Summary of RNAi screens identifying ZIP-2 and CEBP-2 as essential transcriptional regulators of light-induced *cyp-14A5*p::GFP expression. (**C**) Representative epifluorescence images showing light-induced *cyp-14A5*p::GFP expression in wild type, *zip-2*, and *cebp-2* loss-of-function mutants. Scale bar: 50 μm.

*Figure 2 continued on next page*

*Figure 2 continued*

(**D**) Representative epifluorescence images showing light-induced *cyp-14A5p::GFP* expression and no light-induced *irg-1p::GFP* expression in wild type animals. Scale bar: 50 μm. (**E**) Representative epifluorescence images showing light-induced *cyp-14A5p::GFP* expression unaffected by a UV shield. Scale bar: 50 μm. (**F**) Representative epifluorescence images showing light-induced *cyp-14A5p::GFP* expression by red, green, blue LED light sources of equal intensities. Scale bar: 50 μm. (**G**) Quantification of (**E**) and (**F**). ***p<0.001, n.s., non-significant. (**H**) Representative epifluorescence images showing light-induced *cyp-14A5p::GFP* expression unaffected in *rpl-28p::zip-2uORF* transgenic animals. Scale bar: 50 μm. (**I**) Representative epifluorescence images showing light-induced *cyp-14A5p::GFP* expression abolished in *eif-2alpha* mutants. Scale bar: 50 μm. (**J**) Schematic model for light-induced transition of *zip-2* mRNA from translating uORF to mORF, leading to increased ZIP-2 and subsequent increased transcriptional *cyp-14A5* expression in cooperation with CEBP-2. Other molecular players are omitted, and uORF and mORF are separated for clarity.

*Estes et al., 2010*; *Kniazeva and Ruvkun, 2025*; *Reddy et al., 2016*), our results suggest the specific roles of ZIP-2 in mediating light-induced *cyp-14A5* but not *irg-1* reporter expression, suggesting the involvement of additional stress context-specific factors in these processes.

We further explored conditions and mechanisms leading to light-induced *cyp-14A5p::GFP*. To test potential effects of ultraviolet (UV) irradiation from our visible light LED, we used a UV-masking shield to block UV irradiation. However, this did not affect visible LED light-induced *cyp-14A5p::GFP* expression (*Figure 2E*). In addition, we found that the LED light exposure of equal intensities (1500 Lux, 24 hrs) but at different wavelengths (red, green, blue) led to differential *cyp-14A5p::GFP* expression (*Figure 2F and G*), showing stronger effects of shorter wavelengths in the visible-light spectrum.

A pseudo-open reading frame (uORF) in the 5′ untranslated region (UTR) of *zip-2* mRNA inhibits the ribosomal translation of the ZIP-2 main open reading frame (mORF) in the context of PA14 pathogen exposure (*Dunbar et al., 2012*). However, constitutive expression of *zip-2* uORF by the *rpl-28* promoter did not affect light-induced *cyp-14A5p::GFP* (*Figure 2H*). Furthermore, a CRISPR phospho-site knock-in mutation of *eif-2alpha(S49A)* did not affect global translation (*Ma et al., 2023*), yet abolished light-induced *cyp-14A5p::GFP* (*Figure 2I*). As the eukaryotic eIF2alpha complex facilitates translational switch from uORF to mORF upon stress-induced ribosomal stall at uORF (*Brito Querido et al., 2024*; *Costa-Mattioli and Walter, 2020*; *Mir et al., 2024*), these results suggest that it is the *zip-2* uORF translational inhibition, not the uORF protein product function, that mediates visible light-increased ZIP-2 translation and subsequent *cyp-14A5p::GFP* expression (*Figure 2J*).

## Light-induced CYP-14A5 enhances behavioral memory

Although EIF-2alpha and ZIP-2/CEBP-2 functions appear essential for light-induced up-regulation of *cyp-14A5*, the *zip-2* or *cyp-14A5* loss-of-function null mutants show no apparent body-size, morphological, feeding, defecation, or developmental defects under dark or LED light treatment (1500 Lux for 16 or 24 hrs) conditions (*Figure 3—figure supplement 1*). These data suggest that transient visible light exposure or light-induced *cyp-14A5* activation by ZIP-2 does not broadly impact development, aging, or physiology, unlike long-term visible light exposure, which has been shown to robustly shorten lifespans in *C. elegans* (*De Magalhaes Filho et al., 2018*).

The lack of obvious morphological and basal behavioral defects led us to explore whether light exposure influences other aspects of *C. elegans* biology, particularly behavioral plasticity and associative memory formation that might require integration of body physiology. Specifically, we chose a learning paradigm in which animals learn to avoid an innately attractive odor butanone after it is paired with aversive stimuli (*Chandra et al., 2023*; *Kauffman et al., 2010*). *C. elegans* can consolidate this learning into a long-lasting memory for up to 16 hrs once the repetitive training is followed by sleep and recovery post learning (*Chandra et al., 2023*). Using this conditioning protocol (*Figure 3A*), we observed that animals exposed to ambient light (approximately 500–1000 Lux) during olfactory associative learning and recovery exhibited significantly enhanced memory retention compared to those maintained in darkness (*Figure 3B*). To pinpoint the critical period for light exposure, we deprived animals of light in two-hour intervals immediately post learning. Interestingly, light deprivation during the first 2–4 hrs post learning resulted in markedly impaired memory retention (*Figure 3C*). These results suggest that environmental light exposure enhances aversive cue association post learning and is not required for learning itself but is required post learning concurrent with sleep for consolidation of memory (*Chandra et al., 2023*).

Does the light-modulated behavioral memory consolidation require light activation of the ZIP-2 pathway? To address this question, we examined the behavioral memory of two independent *zip-2*

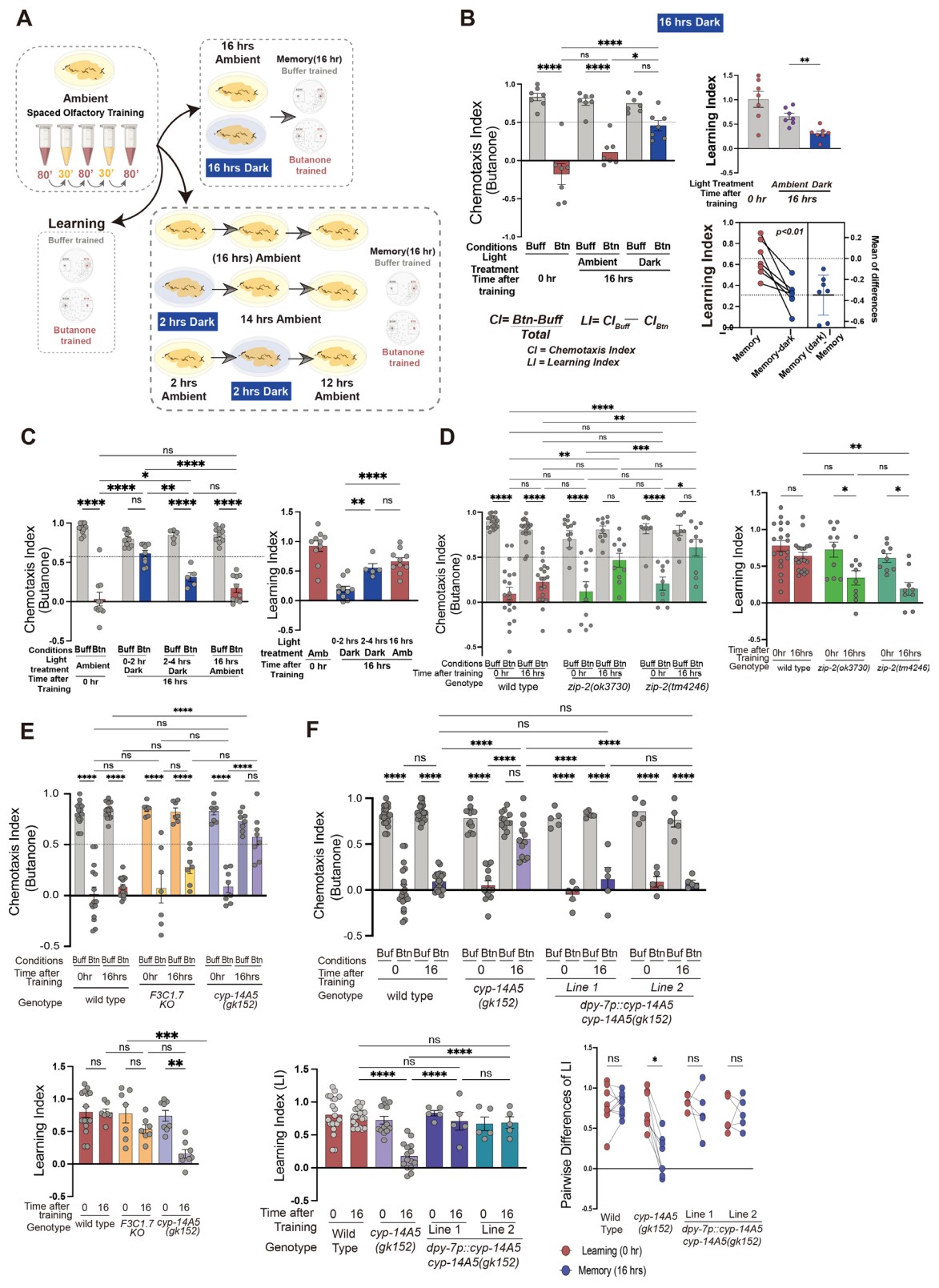

**Figure 3.** Light promotes behavioral plasticity and memory consolidation via a ZIP-2/CYP axis in hypodermis. (**A**) Schematic of behavioral setup to test effects of dark/light on olfactory memory. (**B**) Wild-type learning and memory after 16 hours of dark exposure. 7 trials, 50–200 animals per trial/condition. Two-way ANOVA shows significant differences in chemotaxis (CI) under ambient light conditions (approximately 600 Lux) but not when the assay plates were placed in the dark. Learning (LI) indices reflect the differences between buffer- and butanone-treated animals, attenuated under dark (one-way

*Figure 3 continued on next page*

*Figure 3 continued*

ANOVA). Pairwise *t*-tests of the amount of memory retention under light and dark recovery reveal the degree of memory loss under dark. (**C**) Dark exposure timeline shows that exposing animals to dark immediately after training (0–2 hr, Dark) hampered memory retention, whereas dark conditions for the 2–4 hr period are less sufficient to induce memory loss (two-way ANOVA). The lack of differences between buffer and butanone-trained animals is reflected in the respective LIs (one-way ANOVA). (**D**) Two different loss-of-function alleles of *zip-2(tm4246)* or *zip-2(ok3730)* both showed impaired memory, but learning remained intact (two-way ANOVA). 5–10 trials, 50–200 animals per trial/condition. (**E**) Memory impairment of *cyp-14A5(gk152)* but not *F43C1.7* null mutant animals (two-way ANOVA for CIs) and one-way ANOVA for LIs. 7–14 trials, 50–200 animals per trial/condition. (**F**) Memory defects of *cyp-14A5(gk152)* mutants are rescued by hypodermal expression of wild-type *cyp-14A5*. Two independent transgenic lines show similar results with comparison of pairwise differences in LIs and the amount of memory rescued by hypodermal *cyp-14A5* (Cis: two-way ANOVA; Lis: one-way ANOVA). *$p<0.05$, **$p<0.01$, ***$p<0.001$, ****$p<0.0001$, n.s., non-significant.

The online version of this article includes the following figure supplement(s) for figure 3:

**Figure supplement 1.** No apparent effects of short-term transient visible LED light exposure on development, morphology, simple behaviors, and lifespans.

---

deletion mutants as compared to wild type (*cebp-2* mutants are pleiotropically sick and thus not included). We found that both the *ok3730* and *tm4246* deletion mutations of *zip-2* caused significantly impaired memory consolidation but not learning (**Figure 3D**). Strikingly, the loss-of-function mutation of *cyp-14A5*, but not *F43C1.7* (another ZIP-2 target gene induced by visible light), also impaired memory retention as *zip-2* (**Figure 3E**), indicating a crucial role of the ZIP-2/CYP-14A5 regulatory axis in mediating light-modulated memory consolidation. To further delineate the role of CYP-14A5, we performed tissue-specific rescue experiments in the behavioral memory assay. We found that hypoderm-specific expression of *cyp-14A5* restored the behavioral memory in the *cyp-14A5* mutant (**Figure 3F**). We observed similar degrees of rescue by two independently derived lines expressing hypoderm-specific *dpy-7p::cyp-14A5* transgenes. These findings strongly suggest that hypodermal induction of CYP-14A5 by ZIP-2 plays a central role in mediating light-modulated behavioral memory.

## The *cyp-14A5* promoter as a versatile tool for light-inducible gene expression

The light-responsive nature of the *cyp-14A5* promoter prompted us to explore its potential as a tool for controlling gene expression. We generated synthetic constructs driving the expression of diverse effectors under the *cyp-14A5* promoter to confer striking organismal phenotypes. We previously found that *zip-10* expression promotes stress-induced organismal death, i.e. phenoptosis (**Pandey and Ma, 2022**; **Wang et al., 2023**). Driven by the *cyp-14A5* promoter, light-induced *zip-10* expression indeed caused a robust light-dependent rapid aging or phenoptosis-like phenotype with markedly shortened median and maximal lifespans (**Figure 4A–D**). We confirmed that LED light exposure markedly induced *zip-10* expression, as evidenced by robust ZIP-10-tagging mCherry fluorescence in major non-neuronal tissues of transgenic animals, only after light (1500 Lux, 24 or 48 hrs) exposure (**Figure 4C**).

To test organismal behavioral outcomes, we expressed *nlp-22* under the *cyp-14A5* promoter. *nlp-22* was previously identified as a sleep-promoting neuropeptide (**Bringmann, 2018**; **Nelson et al., 2013**; **Van der Auwera et al., 2020**), overexpression of which can cause drastic reduction of pumping and locomotion speed, characteristic of sleep behaviors in *C. elegans*. We found that *cyp-14A5p::nlp-22* can indeed trigger striking behavioral quiescence upon light exposure (1500 Lux, 24 or 48 hrs), as quantified by pumping rates, bending frequencies, and locomotion speed (**Figure 4E–G**). Quantitative analysis by WormLab reveals that the behavioral quiescence induced by *nlp-22* corresponded to characteristic bouts of sleep (**Figure 4—figure supplement 1**). These proof-of-concept studies demonstrate that the *cyp-14A5* promoter enables light-dependent ectopic induction of gene expression, offering a flexible tool for probing gene function, studies of organismal biology, behaviors, and synthetic physiology applications.

## Discussion

Our findings uncover a previously unknown light-induced transcriptional pathway in *C. elegans* that operates independently of known visual light receptors. Our study also establishes a functional link between ambient light and behavioral plasticity through a ZIP-2/CYP regulatory axis. The discoveries

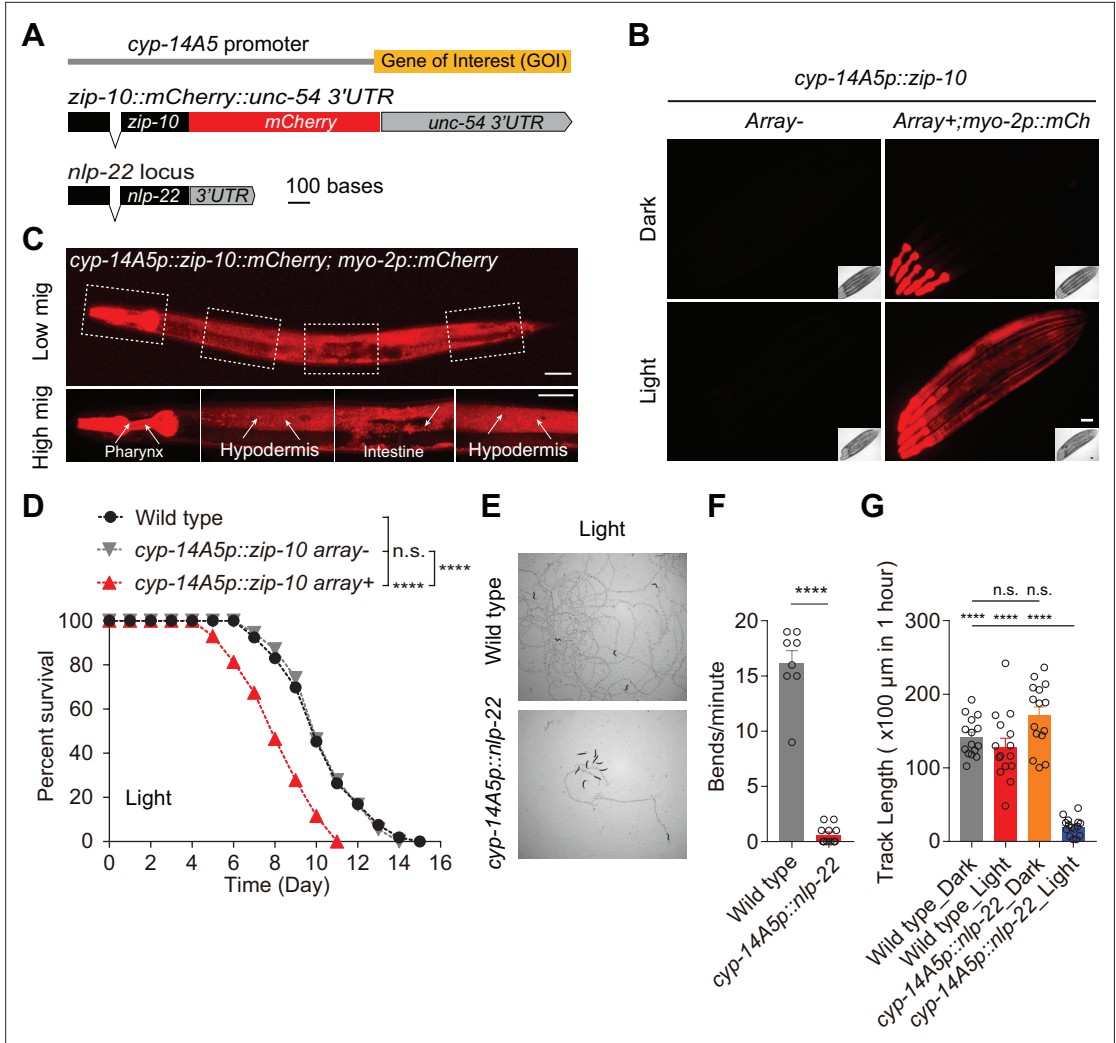

**Figure 4.** Light-induced gene expression drives organismal phenotypes, including sleep and shortened lifespans. (**A**) Schematic of synthetic constructs for light-inducible *nlp-22* and *zip-10::mCherry* using the *cyp-14A5* promoter. (**B**) Representative compound epifluorescence images showing light-induced *cyp-14A5p::zip-10::mCherry* activation. Scale bar: 50 μm. Pharyngeal muscle-specific *myo-2p::mCherry* was used as a co-injection marker. (**C**) Representative confocal fluorescence images showing light-induced *cyp-14A5p::zip-10::mCherry* activation (1500 Lux, 24 hrs starting at 24 hrs post L4) in major non-neuronal tissues (hypoderm, intestinal cells indicated by arrows). Scale bar: 50 μm. (**D**) Representative lifespan curves showing that light-induced *zip-10* can markedly shorten lifespan through transient light exposure (1500 Lux, 24 hrs starting at 24 hrs post L4). Survival was analyzed using the Kaplan–Meier method with log-rank test. ****p<0.0001. (**E**) Representative bright field images showing quiescent sleep behaviors by light-induced *cyp-14A5p::nlp-22* expression through transient light exposure (1500 Lux, 24 hrs starting at 24 hrs post L4). 10 worms were analyzed per experiment, with three independent biological replicates. (**F**) Quantification of population bending frequencies for transient light-treated (1500 Lux, 24 hrs starting at 24 hrs post L4) control wild type and *cyp-14A5p::nlp-22* animals (Left to right, *n* = 8, 11; Student's two-tailed unpaired *t*-test). (**G**) Quantification of population track lengths for control wild type and *cyp-14A5p::nlp-22* animals with transient light (1500 Lux, 24 hrs starting at 24 hrs post L4) or darkness treatments (Left to right, *n* = 15, 13, 12, 13; one-way ANOVA with Tukey's multiple comparison test). ***p<0.001, ****p<0.0001, n.s., non-significant.

The online version of this article includes the following figure supplement(s) for figure 4:

**Figure supplement 1.** WormLab analysis reveals sleep bouts caused by light-induced *cyp-14A5p::nlp-22* expression.

of this pathway and its organismal functions open potential avenues for understanding body-brain communication and how environmental cues can shape physiological and behavioral processes.

The specificity of this pathway is particularly intriguing, as *cyp-14A5* is robustly induced by light at wavelength and intensity that do not apparently alter ambient temperature (**Figure 1—figure supplement 1**). Previous studies identified *cyp-14A5* as one of many genes moderately regulated by bacterial pathogens (**Sinha et al., 2012**; **Troemel et al., 2006**; **Wong et al., 2007**), yet the classic pathogen-inducible gene reporter *irg-1p::GFP* was not apparently activated by light conditions that

induced *cyp-14A5p::GFP*. We found that heat-killed bacteria or starvation suppressed light induction of *cyp-14A5p::GFP*, suggesting a likely modulatory mechanism from bacteria. These observations raise important questions about how ZIP-2 and *cyp-14A5* are selectively activated by light, directly or indirectly via bacteria. Given the crucial roles of ZIP-2 and eIF2alpha we discovered for light-induced expression of *cyp-14A5* and the established role of the uORF at the 5' untranslated region of *zip-2* RNA in ZIP-2 regulation (*Dunbar et al., 2012*), it is plausible that light may regulate ZIP-2 translation by *zip-2* RNA photo-oxidation at specific sites, eIF2alpha phosphorylation, and specialized ribosomal signaling (*D'Orazio and Green, 2021*; *Genuth and Barna, 2018*; *Sinha et al., 2024*). Of note, bacterial chromophores (*Elahi and Baker, 2024*), including porphyrins, flavins, or indole metabolites, can generate reactive oxygen or oxidation products upon illumination, thereby possibly producing exogenous oxidative cues that induce *cyp-14A5* in *C. elegans*. Further investigations into the upstream signaling events, the molecular sensors linking light exposure to ZIP-2/CEBP-2 activation, and how live bacteria modulate light-induced ZIP-2/*cyp-14A5* are warranted.

Our behavioral assays demonstrate that light exposure enhances olfactory associative memory, providing direct evidence for the functional relevance of light-induced *cyp-14A5* expression. Interestingly, light exposure appears to exert its effects during a specific temporal window hours following learning, suggesting that light-induced transcriptional changes post learning play a key role in memory consolidation. The discovery of *cyp-14A5* as a key effector in this pathway also provides new insights into how non-neuronal tissues contribute to behavioral plasticity. Our findings suggest that light-induced CYP-14A5 and CYP-dependent metabolic or signaling changes in the hypoderm may communicate with the nervous system to influence behavioral memory. Importantly, the moderate light intensities used in our behavioral assays are sufficient to trigger measurable CYP-14A5 induction without eliciting the stress responses observed under higher light levels. This suggests that physiologically tuned, intermediate light exposure engages a systemic metabolic program that supports memory stabilization. This body-brain communication axis highlights the importance of systemic integration in mediating complex physiological and behavioral responses to environmental cues (*Aghayeva et al., 2021*; *Fukuda et al., 2025*; *Liu et al., 2022*; *Zhang et al., 2018*).

Beyond its biological significance, the light-inducible *cyp-14A5* promoter offers a useful new tool for gene expression studies in *C. elegans*. The ability to drive ectopic gene expression in response to light provides a versatile system for temporally controlled genetic manipulations. Our demonstration of light-induced sleep and mortality phenotypes through ectopic gene expression illustrates the potential applications of this tool in studying diverse biological processes in synthetic biology and physiology. Previous studies have elegantly used heat shock or drug-inducible promoters for temporally controlled gene expression in *C. elegans* (*Monsalve et al., 2019*; *Stringham et al., 1992*; *Wei et al., 2012*). The light-inducible *cyp-14A5* promoter provides an alternative, simple-to-implement approach and might be particularly useful when the drug-inducible system is cumbersome, or heat shock effects are undesirable. Moreover, the combination of single-copy integrated and validated *cyp-14A5p::cGAL* (see Method) with modular *UASp::effector* constructs (*Wang et al., 2017*) further expands the flexibility and generalizability of this system, enabling researchers to readily plug in diverse effectors for broad applications across cell types and pathways.

While our study uncovers a novel light-responding mechanism with functional consequences in *C. elegans*, several limitations exist. First, the precise molecular mechanism by which visible light activates ZIP-2 and/or CEBP-2 remains unclear, as does the upstream signaling cascade linking light exposure to transcriptional activation. Second, although we demonstrate a functional connection between light-induced *cyp-14A5* expression and behavioral outcomes, the exact molecular interplay between peripheral transcriptional changes and neural plasticity requires further exploration. Finally, while the *cyp-14A5* promoter serves as a useful genetic tool, it does not confer tissue specificity, and its ectopic effector expression requires control for light effects. These limitations provide fertile ground for future research to build upon our findings.

## Materials and methods
### *C. elegans* strains

*C. elegans* strains were grown on nematode growth media (NGM) plates seeded with *Escherichia coli* OP50 at 20°C with laboratory standard procedures unless otherwise specified. The N2 Bristol

strain was used as the reference wild type (*Brenner, 1974*). Mutants and integrated transgenes were back-crossed at least five times. Genotypes of strains used are as follows: *dmaIs156 IV [cyp-14A5p:: cyp-14A5::GFP; unc-54p::mCherry], agIs17 IV [irg-1p::gfp], dmaEx [dpy-7p::cyp-14A5; myo-2p::mCherry], dmaEx [cyp-14A5p::zip-10::mCherry; myo-2p::mCherry], dmaEx [cyp-14A5p::nlp-22; myo-2p::mCherry], cebp-2 (tm5421) I, eif-2alpha(rog3) I, zip-2(tm4248) III, zip-2(ok3730) III, cyp-14A5(gk152) V.* Single-copy integration strains based on the UAS-cGAL system (*Wang et al., 2017*): *jsTi1453 jsSi1518 [UAS 11X Δpes10 GFP-C1] I; jsSi1988 tanSi331 [pHW2186 cyp-14A5p::cGAL-tbb-2 3'UTR; cup-4p::mScarlet::tbb-2 3'UTR] IV.*

PCR fusion constructs were used to generate transgenes (*Hobert, 2002*), using primer sequences:

DM1310_*cyp-14A5*Pro c5p TCAACCACATCTTCCGATCA;
DM1311_*cyp-14A5*Pro to GFP c3p
CGACCTGCAGGCATGCAAGCTgatctttgttggacagaatagtttt;
DM2857_*dpy-7*p to *cyp-14A5* codutr Forward TGTCTCTGACGCCTGTGAGT;
DM2858_*dpy-7*p to *cyp-14A5* codutr Reverse
GATAAAGCAACGATGAAAACGCTCATTTTGTTTTCACAGAGCGGTAGA;
DM2944_*zip-2*uORF to *rpl-28*p-GOI-mCherry-5utr fusion F
CATCATAAAATAATTTATTTCCAGGTAAAATGTATCACGCAAAGACAACCACCG;
DM2945_*zip-2*uORF to *rpl-28*p-GOI-mCherry-5utr fusion
catgttatcttcttcaccctttgaggagccAAGCTCCCGTGGGAAGCTTGTG;
DM2948_*zip-10* to *cyp-14A5*p-GOI-mCherry-5utr fusion F
aaaactattctgtccaacaaagatcaaaATGACAACAATGACTAATTCTCTTATTTC;
DM2949_*zip-10* to *cyp-14A5*p-GOI-mCherry-5utr fusion R
catgttatcttcttcaccctttgaggagccGGAATGGTTGATTTGATTATTGAGTTG
DM2952_*nlp-22*cod::3utr to *cyp-14A5*p-GOI fusion
aaaactattctgtccaacaaagatcaaaATGCGTTCCATAATCGTCTTCATCG;
DM2953_*nlp-22*cod::3utr to *cyp-14A5*p-GOI fusion R cggttccactttctcatgagt

## Fluorescence microscopy and imaging

SPE confocal (Leica) and epifluorescence microscopes were used to capture fluorescence images. Animals were randomly picked at the same stage and treated with 1 mM levamisole in M9 solution (31742–250MG, Sigma-Aldrich), aligned on a 2% agar pad on a slide for imaging. Identical setting and conditions were used to compare experimental groups with control. For quantification of GFP fluorescence, animals were outlined and quantified by measuring gray values using the ImageJ software. The data were plotted and analyzed by using GraphPad Prism 10.

For light-induced reporter imaging, reporter animals (synchronized young adults, 24 hrs post L4) were exposed to white light (1500 Lux for 16 or 24 hrs, by Viribright 12-Watt, 800 Lumen, LED Desk Lamp Dimmable Office Lamp). For blue, green (SPE confocal, Leica), and red light (HQRP 660 nm 14 W LED pure red) conditions, animals of the same stage were exposed to the same intensities (1500 Lux, 16 or 24 hrs). Control groups from the same batch of animals were maintained in darkness by opaque shields. Light intensities and temperature were quantitatively measured by digital light meters and thermometers.

## RNA sequencing

A synchronized population of wild-type young adult (24 hrs post L4) animals was exposed to LED light (1500 lux, 24 hr duration, Viribright 12-Watt, 800 Lumen, LED Desk Lamp Dimmable Office Lamp). Control groups from the same batch of animals were maintained in darkness by opaque light shields. Four independent biological replicates were used for both light-treated and control groups. For sample collection, the animals were washed down from NGM plates using M9 solution and bacteria-cleaned with M9 washing in centrifuge tubes, homogenized by tissue disruptors, and subjected to RNA extraction using the RNeasy Mini Kit from Qiagen. 1 mg total RNA from each sample was used for sequencing library construction. The libraries were constructed and sequenced for paired-end 150 bp by DNBseq (Innomics). The cleaned RNAseq reads were mapped to the genome sequence of *C. elegans* using hisat2, and the mapped reads were assigned to the genes using featureCounts (*Kim et al., 2015*; *Liao et al., 2014*). The abundance of genes was expressed as RPKM (reads per kilobase

per million mapped reads) and identification of differentially expressed genes (*Supplementary file 1*) was performed using the DESeq2 package (*Love et al., 2014*).

### *C. elegans* behavioral assays

The olfactory behavioral memory assay was as described previously with modification (*Chandra et al., 2023*; *Kauffman et al., 2010*). Briefly, 1-day-old adult worms (24 hrs post L4) were washed with S basal buffer (0.1 M NaCl, 0.05 M $K_3PO_4$, pH 6.0) off 10 cm NGM plates and into microfuge tubes, where they were washed three times with S basal buffer. The animals were split into two groups; one group was added to a microfuge tube of S basal media and the other group was added to a micro-fuge tube of 1:10,000 dilution of butanone in S basal. The microfuge tubes were then rotated for 80 minutes. The odor training includes three 80-minute cycles of training with odor, or a control buffer interspersed with two 30-minute periods of feeding with OP50 *E. coli* bacteria. For the chemotaxis assay, 1 µL of (1 M) $NaN_3$ was pipetted onto the odor and diluent spots in 10 cm plastic Petri dishes. 1 µL of 200 proof ethanol was added to the diluent spot and 1 µL of 1:1000 butanone was added to the odor spot, while S basal or butanone-trained worms were dropped onto the middle of 10 cm plastic Petri dishes. The recovery period was either under darkness or ambient light (approximately 600 Lux) for 16 hrs or was kept under darkness for 2 or 4 hr periods followed by light exposure and chemotaxis to assay behavioral memory.

For sleep analysis induced by *cyp-14A5p::nlp-22*, the bending angles, moving average speed, and track length of *C. elegans* after 48 hr of exposure to either light or dark conditions were analyzed using WormLab. In such an experiment, a synchronized population of young adult (24 hrs post L4) animals of indicated genotype or transgene expression was used. After light exposure or control dark treatment, they were transferred to a fresh NGM plate seeded with a small OP50 bacterial lawn and allowed to settle for at least ten minutes to recover at room temperature. After the recovery period, a 1-hr recording session was conducted using WormLab. Bending angles were calculated as described in the referenced method as a metric for sleep behaviors (*Chandra et al., 2023*). Moving average speed was determined by tracking the displacement of the worms over time.

### Statistics

Numerical data were analyzed using GraphPad Prism 10 Software (Graphpad, San Diego, CA) and presented as means ± S.D. unless otherwise specified, with p values calculated by unpaired two-tailed *t*-tests (comparisons between two groups), one-way ANOVA (comparisons across more than two groups) and two-way ANOVA (interaction between genotype and treatment), with post hoc Tukey and Bonferroni's corrections. The lifespan assay was plotted and quantified using Kaplan–Meier lifespan analysis, and p values were calculated using the log-rank test.

## Acknowledgements

Some strains were provided by the Caenorhabditis Genetics Center (CGC), which is funded by the NIH Office of Research Infrastructure Programs (P40 OD010440), and by Dr. E Troemel. We also thank the *C. elegans* Reverse Genetics Core Facility (University of British Columbia), National Bioresource Project (S Mitani, Tokyo Women's Medical University, Tokyo, Japan), https://wormbase.org/ (NIH grant #U24 HG002223 to P Sternberg), https://wormbase.org/ (NIH grant #OD010943 to DH Hall), Aging Atlas (Dr. M Wang), and CenGen (cengen.org) for the resources. The work was supported by NIH grants (R35GM139618 to DKM), BARI Investigator Award (DKM), and UCSF PBBR New Frontier Research Award (DKM).

## Additional information

### Funding

| Funder | Grant reference number | Author |
| --- | --- | --- |
| National Institute of General Medical Sciences | R35GM139618 | Dengke K Ma |

| Funder | Grant reference number | Author |
|---|---|---|
| BARI Investigator Award | | Dengke K Ma |
| UCSF PBBR New Frontier Research Award | | Dengke K Ma |
| NIH Office of Research Infrastructure Programs | P40 OD010440 | |

The funders had no role in study design, data collection and interpretation, or the decision to submit the work for publication.

## Author contributions

Zhijian Ji, Data curation, Formal analysis, Investigation, Writing – original draft, Writing – review and editing; Bingying Wang, Rashmi Chandra, Junqiang Liu, Data curation, Formal analysis, Investigation; Supeng Yang, Michael Egan, Data curation; Yong Long, Data curation, Formal analysis; Fujia Han, Han Wang, Investigation; Noelle L'Etoile, Resources, Supervision, Writing – review and editing; Dengke K Ma, Conceptualization, Resources, Data curation, Supervision, Funding acquisition, Investigation, Writing – original draft, Project administration, Writing – review and editing

## Author ORCIDs
Zhijian Ji ⓘ https://orcid.org/0009-0005-6314-0961
Junqiang Liu ⓘ https://orcid.org/0000-0002-2953-487X
Fujia Han ⓘ https://orcid.org/0009-0009-7079-6108
Han Wang ⓘ https://orcid.org/0000-0002-1933-5762
Dengke K Ma ⓘ https://orcid.org/0000-0002-5619-7485

Reviewer #1 (Public review): https://doi.org/10.7554/eLife.108507.3.sa1
Reviewer #2 (Public review): https://doi.org/10.7554/eLife.108507.3.sa2
Reviewer #3 (Public review): https://doi.org/10.7554/eLife.108507.3.sa3
Author response https://doi.org/10.7554/eLife.108507.3.sa4

# Additional files

## Supplementary files
MDAR checklist

Supplementary file 1. RNA-seq Dataset_Light Treatment vs Dark Control.

## Data availability
All data generated or analyzed during this study are included in the manuscript and supporting files.

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
