## [Editor Report · eLife Assessment]

This **important** study uncovers a previously unrecognized light-responsive pathway in *C. elegans* that depends on live food bacteria and is mediated by the bZIP factors ZIP-2/CEBP-2 and the cytochrome P450 enzyme, CYP-14A5. The authors show that this bacteria-linked pathway modulates long-term memory and can be harnessed as a low-cost light-inducible expression system, opening new directions for sensory biology and genetic engineering in worms. The exact means by which live bacteria modulate light signal that activates ZIP-2/CEBP-2 in the worm remains to be elucidated. The evidence supporting the pathway's role uses multiple genetic, transcriptional, and behavioral assays and is **convincing**.

---

## [Referee Report · Reviewer #1 (Public review)]

Summary:

The authors set out to understand how animals respond to visible light in an animal without eyes. To do so they used the *C. elegans* model, which lacks eyes, but nonetheless exhibits robust responses to visible light at several wavelengths. Here, the authors report a promoter that is activated by visible light and independent of known pathways of light resposnes.

Strengths:

The authors convincingly demonstrate that visible light activates the expression of the cyp-14A5 promoter driven gene expression in a variety of contexts and report the finding that this pathway is activated via the ZIP-2 transcriptionally regulated signaling pathway.

Weaknesses:

Because the ZIP-2 pathway has been reported to activated predominantly by changes in the bacterial food source of *C. elegans* -- or exposure of animals to pathogens -- it remains unclear if visible light activates a pathway in *C. elegans* (animals) or if visible light potentially is sensed by the bacteria on the plate which also lack eyes. Specifically, it is possible that the the plates are seeded with excess *E. coli*, that *E. coli* is altered by light in some way and in this context alters its behavior in such a way that activates a known bacterially responsive pathway in the animals. Consistent with this possibility the authors found that heat-killed bacteria prevented the reporter activation in animals. This weakness would not affect the ability to use this novel discovery as a tool, which would still be useful to the field.

---

## [Referee Report · Reviewer #2 (Public review)]

Summary:

Ji, Ma and colleagues report the discovery of a mechanism in *C. elegans* that mediates transcriptional responses to low intensity light stimuli. They find that light-induced transcription requires a pair of bZIP transcription factors and induces expression of a cytochrome P450 effector. This unexpected light-sensing mechanism is required for physiologically relevant gene expression that controls behavioral plasticity. The authors further show that this mechanism can be co-opted to create light-inducible transgenes.

Strengths:

The authors rigorously demonstrate that ambient light stimuli regulate gene expression via a mechanism that requires the bZIP factors ZIP-2 and CEBP-2. Transcriptional responses to light stimuli are measured using transgenes and using measurements of endogenous transcripts. The study shows proper genetic controls for these effects. The study shows that this light-response does not require known photoreceptors, is tuned to specific wavelengths, and is highly unlikely to be an artifact of temperature-sensing. The study further shows that the function of ZIP-2 and CEBP-2 in light-sensing can be distinguished from their previously reporter role in mediating transcriptional responses to pathogenic bacteria. The study includes experiments that demonstrate that regulatory motifs from a known light-response gene can be used to confer light-regulated gene expression, demonstrating sufficiency and suggesting an application of these discoveries in engineering inducible transgenes. Finally, the study shows that ambient light and the transcription factors that transduce it into gene expression changes are required to stabilize a learned olfactory behavior, suggesting a physiological function for this mechanism.

Weaknesses:

The study implies but does not show that the effects of ambient light on stabilizing a learned olfactory behavior are through the described pathway. To show this clearly, the authors should determine whether ambient light has any further effects on learning in mutants lacking CYP-14A5, ZIP-2, or CEBP-2.

---

## [Referee Report · Reviewer #3 (Public review)]

Ji et al. report a novel and interesting light-induced transcriptional response pathway in the eyeless roundworm *Caenorhabditis elegans* that involves a cytochrome P450 family protein (CYP-14A5) and functions independently from previously established photosensory mechanisms. The authors also demonstrate the potential for this pathway to enable robust light-induced control of gene expression and behavior, albeit with some restrictions. Despite the limitations of this tool, including those presented by the authors, it could prove useful for the community. Overall, the evidence supporting the claims of the authors is convincing, and the authors' work suggests numerous interesting lines of future inquiry.

(1) Although the exact mechanisms underlying photoactivation of this pathway remain unclear, light-dependent induction of CYP-14A5 requires bZIP transcription factors ZIP-2 and CEBP-2 that have been previously implicated in worm responses to pathogens. Notably, this light response requires live food bacteria, suggesting a microbial contribution to this phenomenon. The nature of the microbial contribution to the light response is unknown but very interesting.

(2) The authors suggest that light-induced CYP-14A5 activity in the *C. elegans* hypoderm can unexpectedly and cell-non-autonomously contribute to retention of an olfactory memory. How retention of the olfactory memory is enhanced by light generally remains unclear. Additional experiments, including verification of light-dependent changes in CYP-14A5 levels in the olfactory memory behavioral setup, appropriate would help further interpret these otherwise interesting results.

---

## [Author Response]

The following is the authors’ response to the original reviews

**Public Reviews:**

**Reviewer #1 (Public review):**
Summary:The authors set out to understand how animals respond to visible light in an animal without eyes. To do so, they used the *C. elegans* model, which lacks eyes, but nonetheless exhibits robust responses to visible light at several wavelengths. Here, the authors report a promoter that is activated by visible light and independent of known pathways of light responses.Strengths:The authors convincingly demonstrate that visible light activates the expression of the cyp-14A5 promoter-driven gene expression in a variety of contexts and report the finding that this pathway is activated via the ZIP-2 transcriptionally regulated signaling pathway.Weaknesses:Because the ZIP-2 pathway has been reported to be activated predominantly by changes in the bacterial food source of *C. elegans* -- or exposure of animals to pathogens -- it remains unclear if visible light activates a pathway in *C. elegans* (animals) or if visible light potentially is sensed by the bacteria on the plate, which also lack eyes. Specifically, it is possible that the plates are seeded with excess *E. coli*, that *E. coli* is altered by light in some way, and in this context, alters its behavior in such a way that activates a known bacterially responsive pathway in the animals. This weakness would not affect the ability to use this novel discovery as a tool, which would still be useful to the field, but it does leave some questions about the applicability to the original question of how animals sense light in the absence of eyes.

Thank you for the insightful questions and suggestions. We have now performed a key experiment requested. Interesting new data (Fig. S1I) show that light induction of *cyp-14A5p::GFP* requires live bacteria that maintain a non-starved physiological state. Neither plates without food nor plates with heat-killed OP50 support robust induction. We now include this interesting new result in the paper and revised discussion on the bacteria-modulated mechanism but note that this bacterial requirement does not alter the central conclusions of the study. Rather, it reveals an intriguing mechanistic layer, namely, that bacterial metabolic activity likely influences the animal’s sensitivity to environmental light. We are pursuing this host–microbe interaction in a separate study. In the present work, we focus on the intrinsic regulation and functional significance of *cyp-14A5* under standard laboratory conditions with live OP50. Accordingly, we have revised the Results and Discussion to reflect the appropriate scope.

**Reviewer #2 (Public review):**
Summary:Ji, Ma, and colleagues report the discovery of a mechanism in *C. elegans* that mediates transcriptional responses to low-intensity light stimuli. They find that light-induced transcription requires a pair of bZIP transcription factors and induces expression of a cytochrome P450 effector. This unexpected light-sensing mechanism is required for physiologically relevant gene expression that controls behavioral plasticity. The authors further show that this mechanism can be co-opted to create light-inducible transgenes.Strengths:The authors rigorously demonstrate that ambient light stimuli regulate gene expression via a mechanism that requires the bZIP factors ZIP-2 and CEBP-2. Transcriptional responses to light stimuli are measured using transgenes and using measurements of endogenous transcripts. The study shows proper genetic controls for these effects. The study shows that this light-response does not require known photoreceptors, is tuned to specific wavelengths, and is highly unlikely to be an artifact of temperature-sensing. The study further shows that the function of ZIP-2 and CEBP-2 in light-sensing can be distinguished from their previously reported role in mediating transcriptional responses to pathogenic bacteria. The study includes experiments that demonstrate that regulatory motifs from a known light-response gene can be used to confer light-regulated gene expression, demonstrating sufficiency and suggesting an application of these discoveries in engineering inducible transgenes. Finally, the study shows that ambient light and the transcription factors that transduce it into gene expression changes are required to stabilize a learned olfactory behavior, suggesting a physiological function for this mechanism.Weaknesses:The study implies but does not show that the effects of ambient light on stabilizing a learned olfactory behavior are through the described pathway. To show this clearly, the authors should determine whether ambient light has any effect on mutants lacking CYP-14A5, ZIP-2, or CEBP-2. Other minor edits to the text and figures are suggested.

We appreciate the reviewer’s comment. Our study indeed implies that ambient light stabilizes learned olfactory behavior through effects on the described pathway. Importantly, the existing data already address this point. Mutants lacking CYP-14A5, ZIP-2, or CEBP-2 display impaired olfactory memory even when exposed to ambient light, indicating that these genes are required for the behavioral effect of light. Consistent with this, ambient light robustly induces *cyp-14A5p::GFP* in wild-type animals but fails to do so in *zip-2* and *cebp-2* mutants, demonstrating that light-dependent transcriptional activation is blocked upstream in these pathway mutants. Together, these results support the conclusion that ambient light acts through the ZIP-2 → CEBP-2 → CYP-14A5 pathway to stabilize memory. Minor textual and figure revisions have been made where helpful to clarify this point.

**Reviewer #3 (Public review):**
Ji et al. report a novel and interesting light-induced transcriptional response pathway in the eyeless roundworm *Caenorhabditis elegans* that involves a cytochrome P450 family protein (CYP-14A5) and functions independently from previously established photosensory mechanisms. Although the exact mechanisms underlying photoactivation of this pathway remain unclear, light-dependent induction of CYP-14A5 requires bZIP transcription factors ZIP-2 and CEBP-2 that have been previously implicated in worm responses to pathogens. The authors then suggest that light-induced CYP-14A5 activity in the *C. elegans* hypoderm can unexpectedly and cell-non-autonomously contribute to retention of an olfactory memory. Finally, the authors demonstrate the potential for this pathway to enable robust light-induced control of gene expression and behavior, albeit with some restrictions. Overall, the evidence supporting the claims of the authors is convincing, and the authors' work suggests numerous interesting lines of future inquiry.(1) The authors determine that light, but not several other stressors tested (temperature, hypoxia, and food deprivation), can induce transcription of cyp-15A5. The authors use these experiments to suggest the potential specificity of the induction of CYP-14A5 by light. Given the established relationship between light and oxidative stress and the authors' later identification of ZIP-2, testing the effect of an oxidative stressor or pathogen exposure on transcription of cyp-14A5 would further strengthen the validity of this statement and potentially shed some insight into the underlying mechanisms.

We appreciate the reviewer’s thoughtful suggestion. We would like to clarify that the “specificity” we refer to is the strong and preferential induction of *cyp-14A5* by light among pathogen or detoxification-related genes, rather than an assertion that *cyp-14A5* is exclusively light-responsive. This does not preclude the possibility that *cyp-14A5* can also be activated under other conditions. Indeed, prior work from the Troemel laboratory has identified *cyp-14A5* as one of many pathogen-inducible genes, consistent with its role in stress physiology. Our data show that classical pathogen-responsive genes (e.g., *irg-1*) are not induced by light, whereas *cyp-14A5* is strongly induced, highlighting the selective engagement of this cytochrome P450 by light under the conditions tested. We have revised the text to clarify this point.

(2) The authors suggest that short-wavelength light more robustly increases transcription of cyp-14A5 compared to equally intense longer wavelengths (Figure 2F and 2G). Here, however, the authors report intensities in lux of wavelengths tested. Measurements of and reporting the specific spectra of the incident lights and their corresponding irradiances (ideally, in some form of mW/mm2 - see Ward et al., 2008, Edwards et al., 2008, Bhatla and Horvitz, 2015, De Magalhaes Filho et al., 2018, Ghosh et al., 2021, among others, for examples) is critical for appropriate comparisons across wavelengths and facilitates cross-checking with previous studies of *C. elegans* light responses. On a related and more minor note, the authors place an ultraviolet shield in front of a visible light LED to test potential effects of ultraviolet light on transcription of cyp-14A5. A measurement of the spectrum of the visible light LED would help confirm if such an experiment was required. Regardless, the principal conclusions the authors made from these experiments will likely remain unchanged.

Thank you. We have revised the text to clarify this point. “Using controlled light versus dark conditions, we confirmed the finding from an integrated *cyp-14A5p::GFP* reporter and observed its robust widespread GFP expression in many tissues induced by moderate-intensity (500-3000 Lux, 16-48 hr duration) LED light exposure (Fig. 1A). The photometric Lux range is approximately 0.1–0.60 mW/cm^2^ in radiometric (total radiant power) metric given the spectrum of the LED light source.”

(3) The authors report an interesting observation that animals exposed to ambient light (~600 lux) exhibit significantly increased memory retention compared to those maintained in darkness (Figure 4). Furthermore, light deprivation within the first 2-4 hours after learning appears to eliminate the effect of light on memory retention. These processes depend on CYP-14A5, loss of which can be rescued by re-expression of cyp-14A5 in mutant animals using a hypoderm-specific- and non-light-inducible- promoter. Taken together, the authors argue convincingly that hypodermal expression of cyp-14A5 can contribute to the retention of the olfactory memory. More broadly, these experiments suggest that cell-non-autonomous signaling can enhance retention of olfactory memory. How retention of the olfactory memory is enhanced by light generally remains unclear. In addition, the authors' experiments in Figure 1B demonstrate - at least by use of the transcriptional reporter - that light-dependent induction of cyp-14A5 transcription at 500 - 1000 lux is minimal and especially so at short duration exposures. Additional experiments, including verification of light-dependent changes in CYP-14A5 levels in the olfactory memory behavioral setup, would help further interpret these otherwise interesting results.

We thank the reviewer for these thoughtful comments. We agree that understanding how light enhances memory retention at a mechanistic level is an important direction for future work. Regarding the light intensities used in Figure 1B, we would like to clarify that 500–1000 lux does produce a measurable and statistically significant induction of *cyp-14A5p::GFP*, although the magnitude is lower than that observed at higher intensities. We interpret this modest induction as physiologically relevant: intermediate light levels appear sufficient to engage the CYP-14A5–dependent program required for memory stabilization, whereas stronger light intensities are detrimental to learning and reduce behavioral performance. Thus, the behavioral paradigm uses a light regime that activates the pathway without introducing stress-associated confounders.

(4) The experiments in Figure 4 nicely validate the usage of the cyp-14A5 promoter as a potential tool for light-dependent induction of gene expression. Despite the limitations of this tool, including those presented by the authors, it could prove useful for the community.

Thank you and we agree. In addition, we have included in the revised manuscript the single-copy integration strains based on UAS-GAL4 that produced similar results as transgenic strains and will be even more flexible and useful for the community.

**Recommendations for the authors:**

**Reviewing Editor Comments:**
While appreciating the quality and presentation of this important study, we had two major concerns that the authors need to address.(1) Bacteria-versus-worm origin:To rule out a bacterially derived stimulus, we suggest testing whether cyp-14A5p::GFP is inducible without bacteria (or killed bacteria). Checking whether the canonical immune reporters irg-5p::GFP and gst-4p::GFP are also light-inducible will further clarify this point.

We have now performed the key experiment requested by the reviewers. Interesting new data (Fig. S1I) show that light induction of *cyp-14A5p::GFP* requires live bacteria that maintain a non-starved physiological state. Neither plates without food nor plates with heat-killed OP50 support robust induction. Importantly, this requirement does not alter any of the central conclusions of the study. Rather, it reveals an intriguing mechanistic layer, namely, that bacterial metabolic activity influences the animal’s sensitivity to environmental light. We are pursuing this host–microbe interaction in a separate study. In the present work, we focus on the regulation and functional significance of *cyp-14A5* under standard laboratory conditions with live OP50.

We included the data (Fig. 2D) to show that the canonical immune reporter *irg-1p::GFP* is not induced by the light condition that robustly induced *cyp-14A5p::GFP*, and *gst-4p::GFP* is only very mildly induced (Fig. S1J).

(2) Pathway-behaviour link:The behavioural relevance of the newly described pathway is intriguing, but it needs direct support. Ideally, this would require comparing memory in WT, zip-2-/-, cebp-2-/-, and cyp-14A5-/- under both dark and light conditions. But at the very least, it would require testing if constitutive CYP-14A5 rescue in the dark bypasses the requirement of light.

We respectfully submit that additional experiments are not required to support the behavioral conclusions. Our model posits that *cyp-14A5* is required but not sufficient for memory stabilization, one component within a broader set of light-induced genes. Thus, constitutive hypodermal expression of *cyp-14A5* would not be expected to bypass the requirement for ambient light. The existing data are fully consistent with this framework and conclusions of the paper.

**Reviewer #1 (Recommendations for the authors):**
Overall, I think this paper is interesting to the field of *C. elegans* researchers at a minimum, as a light-inducible gene expression system might have a variety of uses throughout the diverse research paradigms that use this model system. With that said, I have a couple of suggestions that I think would substantially impact the ability to interpret these findings, which might be useful for broader implications of the study.(1) Most importantly, the supplemental table of RNA-seq data should likely be updated and discussed further beyond the cyp-14A5 findings. First, the authors report 7,902 genes are differentially expressed in response to light and then break these into upregulated and downregulated genes. But there are only 1,785 upregulated genes and 3,632 downregulated genes. This adds up to 5417 genes, but doesn't match the 7,902 genes reported to change, and I could not find in the text if some other filters were applied that might explain this not adding up.

Thank you for this helpful comment. We agree that the exact numbers depend on statistical thresholds and are therefore somewhat arbitrary. To avoid implying unwarranted precision, we have revised the text to state that “thousands of genes are differentially regulated by light.”

(2) Among the upregulated genes in response to light are irg-5, irg-4, irg-6, irg-8, and gst-4. Indeed, all of these well-studied genes (or most) show even more induction by light than cyp-14A5. It is my opinion that this result needs further criticism as there are existing GFP reporters for gst-4 and irg-5 that are similarly well studied to irg-1, which is in the paper (and is not upregulated). In my opinion, the authors should test if they see activation of the irg-4 and gst-4 GFP reporters by light as well. This would not only validate their RNA-seq but might provide more important evidence for the field, as these other reporters are not considered light-inducible previously. If they are, several major studies might be impacted by this.

Thank you for the comments. We have *irg-1p::GFP* and *gst-4p::GFP* in the lab but did not find other reporters for the genes mentioned from CGC. Neither of the two reporters showed light induction (Figs. 2D and S1J) as strongly as *cyp-14A5p::GFP*. It is possible that *irg-1* and *gst-4* RNA levels are up-regulated but not reflected in our transgenic reporters that used their promoters to drive GFP expression. Stronger light induction of *cyp-14A5p::GFP* is unlikely caused by the multi-copy nature of the transgene since newly generated single-copy integration strains based on the UAS-GAL4 system produced similar robust results for light induction (Fig. S1I and see Method).

(3) Along the same lines, if at least 4 (and likely more) well characterized immune response genes are activated by light and these genes are known to mostly respond to differences in *C. elegans* bacterial food source/diet, then it stands to reason that maybe in this experimental context the light is not acting on "animals" at all, but rather triggering changes in *E. coli* (i.e. changing *E. coli* metabolism or pathogenicity like properties). If true, then perhaps the light affects bacteria in such a way that it activates a previously known bacterial pathogen response mechanism. This should be easy to test by seeing if this reporter is still activated by light in the presence of diverse bacterial diets, which are available from the CGC (CeMBio collection, for example). This is likely very important to the conclusions of the manuscript as it relates to animals sensing light, but might not be as important to the use of this system as a tool.

Thank you for the insightful questions and suggestions. Interesting new data (Fig. S1I) show that light induction of *cyp-14A5p::GFP* requires live bacteria that maintain a non-starved physiological state. Neither plates without food nor plates with heat-killed OP50 support robust induction. Importantly, this requirement does not alter any of the central conclusions of the study. Rather, it reveals an intriguing mechanistic layer, namely, that bacterial metabolic activity influences the animal’s sensitivity to environmental light. We are pursuing this host–microbe interaction in a separate study. In the present work, we focus on the regulation and functional significance of *cyp-14A5* under standard laboratory conditions with live OP50. We have revised the Results and Discussion to reflect the appropriate scope of our study and implications of the new findings.

(4) Lastly, it seems unlikely that nearly half the *C. elegans* genome is transcriptionally regulated by light (or nearly half of the detected genes in the RNA-seq results). It seems likely that this list of 7,902 genes contains false positives. I would suggest upping some sort of filter, like moving to padj < 0.01 instead of 0.05, or adding a 4-fold change filter 2-fold and 0.01 still results in near 5000+ genes changing, which might explain the difference in up and down genes just being due to different padj filters. Along these lines, it is worth noting that the padj is generated using DESeq2 it appears and one of the first assumptions of DESeq2 is that the median expressed genes do not change, and there is a normalization. However, if MOST genes do change in expression, then one of the fundamental assumptions of DESeq2 is not valid, and thus would mean it might not be an appropriate analysis tool - perhaps there is some other normalization that could be done before running DESeq2 due to some other noise present in the RNA-seq runs?

Thank you for this helpful comment. We agree that the exact numbers depend on statistical thresholds and are therefore somewhat arbitrary. To avoid implying unwarranted precision, we have revised the text to state that “thousands of genes are differentially regulated by light.”

(5) Minor point - I would delete the reference to ER in line 92. While most CYPs do localize to the ER, the images shown are not clearly ER and probably do not have enough resolution to make claims about subcellular localization. To me, it would be easier to just delete this claim as it is not required for the main claims of the manuscript.

Reference deleted.

**Reviewer #2 (Recommendations for the authors):**
I have one request for clarification that likely requires additional data. Figure 3 shows that ambient light stabilizes learned changes to chemotaxis and further shows that CYP-14A5 has a similar function. The implication is that light promotes CYP-14A5 expression, which somehow promotes memory consolidation. The authors should test whether memory consolidation in cyp-15A5, zip-2, or cebp-2 mutants is no longer affected by ambient light.

It is also possible to test whether forced expression of CYP14A5 can bypass the effect of 'no light' conditions on memory consolidation.

Thank you for the comments. We respectfully submit that additional experiments are not required to support the behavioral conclusions. Our model posits that *cyp-14A5* is required but not sufficient for memory stabilization, one component within a broader set of light-induced genes. Thus, constitutive hypodermal expression of *cyp-14A5* would not be expected to bypass the requirement for ambient light. The existing data are fully consistent with this framework and conclusions of the paper.

I have several minor suggestions relating to the text and figures.(1) In the introduction, the authors assert that little is known about non-visual light sensing and then list many examples of molecular mechanisms of non-visual light-sensing. They should emphasize that non-visual light sensing is important and accomplished by diverse molecular mechanisms.

Agree and revised accordingly.

(2) Check spacing between gene names (line 109).

Corrected.

(3) There should be a new paragraph break when the uORF experiments are described (line 146).

Corrected.

(4) 'Phenoptosis' is an esoteric word. Please define it (line 206).

Corrected.

(5) 'p' in the transgene name cyp-14A5p::nlp-22 is in italics, unlike the rest of the manuscript.

Corrected.

(6) 'Acknowledgment' should be 'Acknowledgments' (line 384).

Corrected.

(7) The color map in panel 1B should have units.

It was arbitrary unit (now added) to highlight relative not absolute differences.

(8) In panel 1E, it is confusing to have 'DARK' denoted by reddish bars and 'LIGHT' denoted by bluish bars. Perhaps 'DARK' is black/dark grey and 'LIGHT' is white?

Corrected.

(9) In panel 1D, it takes a minute to find the purple diamond. Please mark up the volcano plot to make it easier.

Corrected.

**Reviewer #3 (Recommendations for the authors):**
The authors generally present convincing experiments detailing interesting results in a well-written manuscript.One quick note: the same Bhatla and Horvitz (2015) papers appear to be cited twice [line 52].

Corrected.